# All-In-One Drive: A Comprehensive Perception Dataset with High-Density Long-Range Point Clouds

**Xinshuo Weng, Yunze Man, Jinhyung Park, Ye Yuan, Matthew O'Toole, Kris Kitani**
Robotics Institute, Carnegie Mellon University
{xinshuow, yman, jinhyun1, yyuan2, motoole2, kkitani}@cs.cmu.edu

## Abstract

Developing datasets that cover comprehensive sensors, annotations and out-of-distribution data is important for innovating robust multi-sensor multi-task perception systems in autonomous driving. Though many datasets have been released, they target for different use-cases such as 3D segmentation (SemanticKITTI), radar data (nuScenes), large-scale training and evaluation (Waymo). As a result, we are still in need of a dataset that forms a union of various strengths of existing datasets. To address this challenge, we present the AIODrive dataset, a synthetic large-scale dataset that provides comprehensive sensors, annotations and environmental variations. Specifically, we provide (1) eight sensor modalities (RGB, Stereo, Depth, LiDAR, SPAD-LiDAR, Radar, IMU, GPS), (2) annotations for all mainstream perception tasks (*e.g.*, detection, tracking, prediction, segmentation, depth estimation, etc), and (3) out-of-distribution driving scenarios such as adverse weather and lighting, crowded scenes, high-speed driving, violation of traffic rules, and vehicle crash. In addition to comprehensive data, long-range perception is also important to perception systems as early detection of faraway objects can help prevent collision in high-speed driving scenarios. However, due to the sparsity and limited range of point cloud data in prior datasets, developing and evaluating long-range perception algorithms is not feasible. To address the issue, we provide high-density long-range point clouds for LiDAR and SPAD-LiDAR sensors ($10\times$ than Velodyne-64), to enable research in long-range perception. Our dataset is released and free to use for both research and commercial purpose: http://www.aiodrive.org/.

## 1 Introduction

The present surge towards building autonomous vehicles has undoubtedly advanced computer vision research by generating large diverse datasets acquired from hundreds of hours of data, thousands of hours of manual annotation, and billions of dollars towards the development of a customized sensing platform – the autonomous vehicle. As a result of these investments, large driving datasets [53, 7, 38, 1, 17, 65, 67, 41] have been released to the research community. It is important to note that while these datasets helped to advance perception systems, each dataset has different focuses as shown in Figure 1 (Left). For example, Waymo [53] dataset provides large-scale data for training 3D object detection and tracking algorithms but does not support other perception tasks such as point cloud segmentation. Likewise, Argoverse [8] dataset provides map annotation for improving perception algorithms but cannot be used for algorithms requiring Radar data as provided by nuScenes [7]. To innovate perception systems that require diverse sensor modalities or methods that integrate multiple perception tasks, existing datasets might not be applicable. Also, merging a few existing datasets together is non-trivial because sensor configurations are significantly different across datasets.

As a community, we are in need of a dataset that forms a union of strengths of existing datasets to innovate multi-sensor multi-task perception systems. Also, the perception systems need to be trained and tested against out-of-distribution data to ensure safety. However, building a real-world dataset that combines the strengths of multiple datasets and includes large mount of out-of-distribution data (*e.g.*,

Submitted to the 35th Conference on Neural Information Processing Systems (NeurIPS 2021) Track on Datasets and Benchmarks. Do not distribute.

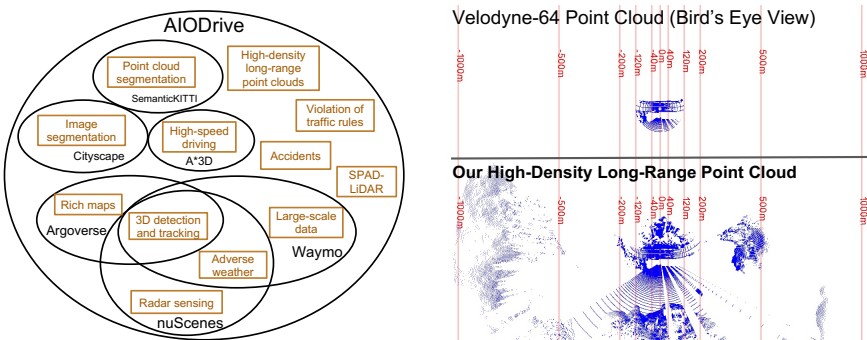

Figure 1: (Left) AIODrive dataset forms a union of various strength of existing datasets, including comprehensive sensors, annotations and out-of-distribution data. (Right) We compare point clouds from Velodyne-64 [26] (about 100k points and a range of 120m) with point clouds from our sensor (about 1M points and a range of 1km), which can be used to innovate long-range perception systems.

car crash) is significantly more challenging and dangerous than building a single-strength dataset without much out-of-distribution data, beyond the capacity of a single research group or university.

One solution that we propose in this work is the use of a simulator, Carla [11], to generate a comprehensive perception dataset, which we call **A**ll-**I**n-**O**ne **D**rive (AIODrive) dataset. Synthetic data generation is able to meet the challenges of creating a comprehensive perception dataset because: (1) a large amount of out-of-distribution data can be safely generated in simulation as the Carla simulator can change the density of traffic, velocity of agents, generate violations of traffic rules, car crashes and change weather and lighting; (2) large amounts of annotation for a multitude of tasks can be automatically generated by combining and post-processing Carla outputs. For example, we can project 2D semantic annotation to 3D given the depth image, resulting in 3D semantic annotation for point clouds. Then, combining with 3D bounding box annotation, 3D semantic annotation can be converted to 3D instance and panoptic segmentation; (3) A 'physical' yet affordable sensing platform can be constructed in simulation to change sensor configuration and even create sensors that are not yet available in public datasets, *e.g.*, long-range high-density LiDAR and SPAD-LiDAR as shown in Figure. 1 (Right), which are only available as early prototype in industry. These powerful sensors can help advance early research in long-range perception before the prototype sensors have been made in product and used in public datasets. To summarize, our AIODrive dataset provides:

(1) 8 sensor modalities: $5\times$ RGB cameras (1 stereo pair); $5\times$ depth cameras, $4\times$ Radar, $3\times$ 1km-range LiDAR at multiple levels of density (up to 1M points), 1km-range SPAD-LiDAR, IMU, and GPS. 4 of the sensors have $360°$ horizontal coverage (camera, LiDAR, SPAD-LiDAR, Radar);

(2) Annotations for all mainstream perception tasks: 2D/3D semantic, instance and panoptic segmentation, 2D/3D bounding boxes, object categories, goals, trajectories, velocity and acceleration;

(3) Diverse environmental variations: adverse weather and lighting, crowded scenes, people running, high-speed driving, violations of the traffic rule, and car crash.

**Domain gap issue.** Though synthetic data generation can be used to create a comprehensive dataset, one might argue that the domain gap between synthetic and real data is a weakness. First, we agree this is the limitation of our dataset. However, we argue that our dataset can still be useful even with this domain gap issue. This argument has been firmly predicated on a body of prior work [46, 34, 44, 18] that has shown, when synthetic data is used correctly, it can be used to enhance perception performance on real data. For example, [34] showed that using synthetic data for augmentation can improve performance for depth prediction on real NYU [50] and SUN RGB-D [51] datasets. [44] showed that using synthetic data created from Unity with free annotation of semantic segmentation can improve segmentation performance on real-world datasets such as KITTI [12], CamVid [5], LabelMe [45], CBCL [2]. Also, [46] showed that augmenting with LiDAR point clouds generated from Carla simulator can improve bird's eye view 2D detection performance on the real-world KITTI dataset. [18] showed that using GTA-V [43] to synthesize LiDAR point clouds for pre-training 3D object detectors can improve $5\%$ average precision on the KITTI dataset. Similar to the success of prior synthetic datasets, we believe that the usefulness of our dataset is also undoubted, as validated by our experiments on real datasets. Again, we emphasize that the role of our dataset is not to replace real datasets. Instead, it can be used in concert with real data, such as using our data to pre-train detectors to improve performance on real data or using our rare driving data as out-of-distribution test data.

The broader impact of our AIODrive dataset is its comprehensive nature allowing for development and evaluation of multi-sensor multi-task perception systems that are not possible with existing datasets. Our dataset includes a super-set of sensors, annotations and environmental variations needed to develop novel perception systems. To provide researchers with various levels of resources access, we have released our dataset for free use. On the other hand, the potential negative impact of our dataset is safety concern. If the data is improperly used, perception systems deployed on real vehicles can cause accidents. To mitigate the potential issue, we provide detailed instructions on our website about how to use the data properly to improve or innovate perception systems.

## 2 Related work

**Perception dataset.** Sensors, environmental variations and annotations are keys to perception datasets. In terms of the annotation, KITTI [12] provides 2D/3D box trajectories, enabling object detection and tracking. To enable image segmentation research, Cityscape [9], Mapillary [35], Apolloscape [55], SYNTHIA [44] datasets are proposed, each having an increased number of annotated frames. For 3D segmentation, SemanticKITTI [1] released point-wise semantic labels on point clouds. As map information such as drivable area is useful in perception, Argoverse [8] manually annotates map semantics to innovate perception algorithm levaring map data.

In addition to annotations, perception datasets also need diverse environmental variations to capture rare driving situations. As prior datasets such as KITTI usually have a small number ($<$10) of agents per frame without complex interactions, H3D [38] was released, with an average of 37 agents per frame to include highly-crowded scenarios with complex agent-agent interactions. To deal with adverse weather and lighting, recent datasets such as CADC [41], nuScenes [7], A*3D[40], Waymo [53] collected data under rainy, snowy, foggy, dusky and night conditions. As prior datasets usually acquired data at a low driving speed (*e.g.*, about 16 km/h in nuScenes), A*3D dataset [40] was proposed to collect data at a much higher speed (*e.g.*, 40-70 km/h).

Regarding the sensing modalities, nuScenes [7] collected the first dataset with Radar data, in addition to standard RGB camera, LiDAR, IMU, and GPS sensors. As earlier datasets collected data in the frontal direction only, ignoring objects to the sides or rear that are also important to decision-making in driving, Argoverse [8], Audi [13], and nuScenes [7] equip their vehicles with multiple LiDAR and camera sensors for $360°$ data capturing.

In comparison to existing datasets with a subset of sensors, annotations and environmental variations, AIODrive provides a super-set of sensors, annotations and environmental variations. Also, beyond standard LiDAR such as Velodyne-64 [26] used in prior datasets for data collection, we provide LiDAR sensors with $10\times$ larger sensing range and 4 levels of point densities, with the highest level having $10\times$ higher point density than Velodyne-64. Importantly, the design of our long-range LiDAR sensors is not imaginary but based on active developments in new LiDAR sensors such as AlphaPrime [27], Ouster [36] and Panasonic [37], which are developed with higher-resolution and longer-range (*e.g.*, 300m) depth sensing. In addition to providing LiDAR sensors, also referred to as APD-LiDAR (avalanche photodiodes), our dataset also provides SPAD-LiDAR (single photon avalanche diode) sensor which records photon counts over space and time. This type of SPAD-LiDAR sensor, although available in industry [47, 6], is not found in public perception datasets for research purpose.

**Synthetic data generation.** Though many existing simulators (*e.g.*, Sim4CV [33], Nvidia Drive [3]) can be used for synthetic data generation, most of these simulators are not open-source (not easy to make modifications) and free-to-use license is not available (*i.e.*, derivative products are not allowed). For the open-sourced simulators, AirSim [48] and Carla [11] are popular due to detailed documentation and diverse sensors. However, AirSim does not allow low-level control over every agent in the way that Carla allows, though AirSim has advantages in aerial data capture. In addition to simulators, commercial video games such as GTA-V [43] can also be used for synthetic data generation but they do not allow low-level control of scene elements. Accordingly, we have selected to use Carla for data generation as it affords the most flexibility and customization.

**Long-range perception.** Increasing the maximum sensing range of perception systems is important for safety in high-speed driving scenarios. However, LiDAR used in existing datasets has limited range, *e.g.*, 120m in KITTI [12], 70m in nuScenes [7], 75m in Waymo [53]. Even with perfect detection accuracy and zero algorithmic latency, a car moving at a speed of 120km/h will only have 3.6 seconds to respond to a detected obstacle with a 120m-range LiDAR. Naturally, enabling perception at a longer-range is preferred for increased safety. To the best of our knowledge, [67] is

Table 1: Comparison of size and sensor modalities. Our dataset has the most comprehensive sensors.

| Dataset | # cities | # hours | # sequences | # annotated images | Stereo | Depth | LiDAR | Radar | SPAD-LiDAR | IMU/GPS | All 360° |
|---|---|---|---|---|---|---|---|---|---|---|---|
| KITTI [12] | 1 | 1.5 | 22 | 15k | ✓ | ✓ | ✓ | | | ✓ | |
| Cityscape [9] | 27 | 2.5 | 0 | 5k | ✓ | | | | | ✓ | |
| Mapillary Vistas [35] | 30 | - | - | 25k | | | | | | | |
| ApolloScape [17, 55] | 4 | - | - | 140k | ✓ | | ✓ | | | ✓ | |
| SYNTHIA [44] | 1 | 2.2 | 4 | 200k | | ✓ | | | | | ✓ |
| H3D [38] | 4 | 0.8 | 160 | 27k | | | ✓ | | | ✓ | |
| SemanticKITTI [1] | 1 | 1.2 | 22 | 43k | | | ✓ | | | | |
| DrivingStereo [52] | - | 5 | 42 | 180k | ✓ | ✓ | ✓ | | | | |
| Argoverse [8] | 2 | 0.6 | 113 | 22k | ✓ | | ✓ | | | ✓ | ✓ |
| EuroCity [4] | **31** | 0.4 | - | 47k | | | | | | | |
| CADC [41] | 1 | 0.6 | 75 | 7k | | | ✓ | | | ✓ | |
| Audi [13] | 3 | 0.3 | 3 | 12k | ✓ | ✓ | ✓ | | | ✓ | ✓ |
| nuScenes [7] | 2 | 5.5 | 1k | 40k | | | ✓ | ✓ | | ✓ | ✓ |
| A*3D [40] | 1 | **55** | - | 39k | ✓ | | ✓ | | | | |
| Waymo Open [53] | 3 | 6.4 | **1150** | 230k | | | ✓ | | | | |
| **Ours** (AIODrive) | 8 | 2.8 | 100 | **100k** | ✓ | ✓ | ✓ | ✓ | ✓ | ✓ | ✓ |

Table 2: Sensor description.

| Sensor | Brief Description |
|---|---|
| 5× RGB Camera | 10Hz frequency, two face forward stereo camera, the others are for left, right and back directions, each with a FoV of 120°, 1920 × 720 |
| 5× Depth Camera | same as the above RGB cameras |
| 3× LiDAR | 64/800/1200 channels, 100k/600k/1M points per frame, 360° horizontal FoV, −90° to 90° vertical FoV, 10Hz frequency, ≤1000m range |
| 1× SPAD-LiDAR | −17° to 18° vertical FoV, 1M points per frame |
| 4× Radar | 10Hz frequency, 360° horizontal FoV with 4 views (left, right, front, back), 150k points per second, ≤1000m range |
| 1× IMU/GPS | 10Hz frequency |

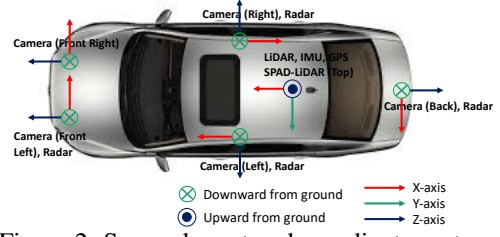

Figure 2: Sensor layout and coordinate systems.

the only work exploring a scenario with up to 300m of depth sensing using three high-resolution RGB cameras. In contrast, our work uses a simulator to collect long-range high-density point clouds. We believe that our data can help aid in the development of long-range perception algorithms before data from real-world long-range sensors become widely available to the research community.

## 3 The AIODrive dataset

### 3.1 Comprehensive sensor suite

To increase robustness to sensor failure, multi-sensor perception approaches [24, 42, 61, 56, 62, 25, 20] are often more favorable than single-sensor approaches [49, 57, 64, 58]. To innovate multi-sensor approach, it is crucial that datasets can provide comprehensive sensing modalities. To that end, we provide common sensors such as RGB, Depth, Stereo camera, LiDAR, IMU and GPS, as well as the Radar and SPAD-LiDAR sensors, which are often not available in prior work as shown in Table 1 (except for nuScenes providing the Radar data). To the best of our knowledge, we are the first to provide the SPAD-LiDAR data in public perception datasets. Also, our camera, LiDAR, Radar and SPAD sensors all have 360° horizontal field of view (FoV).

**Sensor specifications.** We show sensor descriptions in Table 2. Our sensor suite contains five (four for 360° sensing and one for stereo) RGB and five depth cameras, as well as three LiDAR, four Radar, one SPAD-LiDAR and IMU/GPS sensors. All sensors are synchronized with a frequency of 10Hz.

**Sensor layout and coordinate system.** We follow KITTI and use the right-hand rule for coordinate systems. Specifically, for camera/Radar coordinate, we use x axis for the right, y axis pointing downward and z axis for the front direction. For LiDAR and IMU/GPS coordinate, we use x axis for the front, y axis for the left and z axis pointing upward. We summarize sensor layout and coordinate systems in Figure 2. To avoid transforming the coordinate between LiDAR, IMU and GPS sensors, we place these sensors at the same location (on top of the ego-vehicle) in simulator.

**High-density long-range point cloud.** To ensure safety in high-speed driving scenarios, long-range perception [67] is critical. To innovate long-range perception systems, we as a community need public datasets that collect data using longer-range LiDAR sensors than standard 120m-range Velodyne-64 [26]. In anticipation of new high-density long-range LiDAR sensors such as AlphaPrime [27], OS2 [36] and Panasonic [37], we simulate LiDAR sensors with similar specifications to help aid in the development of long-range perception systems. Specifically, we provide three LiDAR sensors, each with a resolution (density) of 100k, 600k, 1M points per frame. Each point in the cloud is a tuple of $(x, y, z, r)$, where $(x, y, z)$ is the 3D location. Also, $r$ is the simulated reflectance (also called intensity) value, which depends on many factors such as the sensor's attenuation factor, distance of the point, and color of the reflection surface. The first LiDAR with 100k points and a range of 120m is to mimic the Velodyne-64, and the other two high-density long-range LiDARs are provided to

Table 3: Comparison of annotation availability. We provide the most complete annotations.

| Dataset | # 2D boxes | # 3D boxes | Trajectory | Image seg. | Point cloud seg. | Motion dynamics | F.g. object class | Map |
|---|---|---|---|---|---|---|---|---|
| KITTI [12] | 80k | 80k | ✓ | | | | | |
| Cityscape [9] | 65k | - | | ✓ | | | | |
| Mapillary Vistas [35] | 200k | - | | ✓ | | | | |
| ApolloScape [17, 55] | 2.5M | 70k | | ✓ | ✓ | | | |
| SYNTHIA [44] | - | - | | ✓ | | | | |
| H3D [38] | - | 1M | ✓ | | | | | |
| SemanticKITTI [1] | - | - | | | ✓ | | | |
| DrivingStereo [52] | - | - | | | | | | |
| Argoverse [8] | - | 993k | ✓ | | | | | ✓ |
| EuroCity [4] | 238k | - | | | | | | |
| CADC [41] | - | 344k | | | | | | |
| Audi [13] | - | 42k | | ✓ | | | | ✓ |
| nuScenes [7] | - | 1.4M | ✓ | | | | | ✓ |
| A*3D [40] | - | 230k | | | | | | |
| Waymo Open [53] | 9.9M | 12M | ✓ | | | | | |
| **Ours** (AIODrive) | 10M | 10M | ✓ | ✓ | ✓ | ✓ | ✓ | ✓ |

Velodyne-64 point cloud      Our dense depth point cloud

Figure 3: **Comparison of point density between Velodyne-64 (left) and our point cloud (right)**. Our point cloud with higher density provides potential for detecting objects at a large distance.

innovate long-range perception systems. All LiDARs are spinning and collecting point clouds via ray-casting. To increase the realism of the LiDAR point clouds, two augmentation mechanisms are used: (1) we randomly drop a small portion of points based on their intensity values, *i.e.*, the lower the intensity is, the higher probability to be dropped; (2) we randomly perturb a small portion of points along the direction of the laser ray, creating noisy distance measurements.

In addition to LiDAR, we generate depth point clouds by projecting five depth images to 3D and then fusion (see supp. for details). Our full-surround depth point cloud has 4M points and 1km range. We show a comparison of Velodyne-64 and depth point cloud in Figure 3. For a car at 130 meters, depth point cloud can capture a decent number of points while Velodyne-64 can not capture any point.

**SPAD-LiDAR** is useful in tasks such as depth sensing [30], non-line-of-sight imaging [31, 16]. In anticipation of next generation SPAD-LiDAR (*e.g.,* ON Semiconductor [47], Leica SPL100 [6]), we simulate SPAD-LiDAR to mimic the configurations of new SPAD-LiDAR sensors that are actively being developed in industry. In comparison to LiDAR (or APD-LiDAR) which requires hundreds of photons received in a short period to trigger an avalanche (*i.e.*, a valid return point), SPAD is designed to measure every single photon. Meanwhile, SPAD-LiDAR is designed to have a higher spatial coverage rate (fill factor), allowing a single laser to get reflected by multiple objects along its propagation path, resulting in multi-echo point clouds. The multi-echo point cloud generated by our SPAD-LiDAR has about 1M points with a sensing range of 1km. Please refer to our supp. for detailed multi-echo SPAD-LiDAR simulation process. Again, we emphasize that our dataset is the first providing SPAD-LiDAR. Please refer to supp. for other sensors such as Radar and depth camera.

### 3.2 Diverse annotations

Annotation availability to various tasks is important to perception datasets. As shown in Table 3, we provide the most comprehensive annotations, which includes 2D-3D box trajectories, image and point cloud segmentation, motion dynamics, fine-grained object class as well as map.

**Bounding box trajectories.** To support 2D-3D detection [57] and re-identification [23], 2D-3D tracking [58], trajectory forecasting [60], we provide 2D-3D box annotations and object identities as shown in Figure 4. Following KITTI [12], we use $(x_1, y_1, x_2, y_2)$ to represent a 2D box, where the $(x_1, y_1)$ and $(x_2, y_2)$ denotes coordinates of the top left and bottom right corners. Truncation and occlusion measurements are also provided. To represent 3D box, we use $(x, y, z, l, w, h, \theta)$, where $(x, y, z)$ is the object center, $(l, w, h)$ denotes the box size and $\theta$ is the heading orientation.

**2D-3D segmentation.** To innovate pixel-level perception algorithms, we provide 2D-3D semantic, instance and panoptic segmentation labels as shown in Figure 5. The 2D segmentation labels are

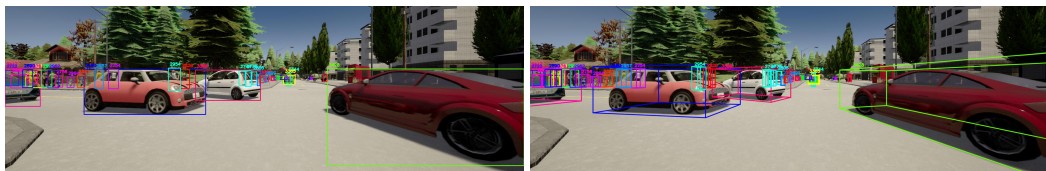

Figure 4: **2D-3D Box Trajectory Annotation.** For each agent, we provide both 2D (left) and 3D (right) tight box annotation, along with a unique ID (visualized with different colors).

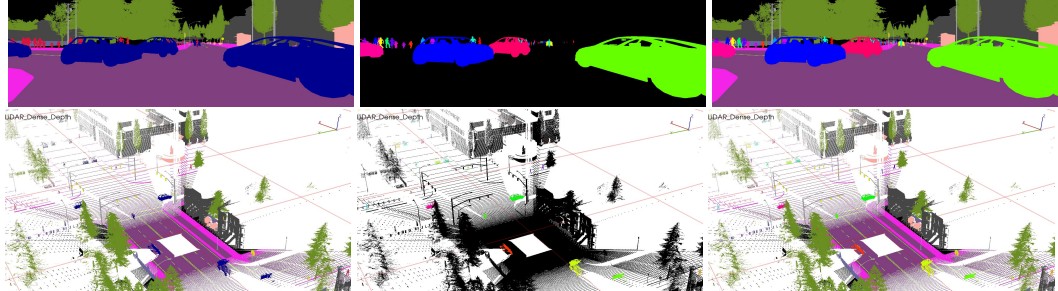

Figure 5: **2D-3D Segmentation Annotation.** We provide both 2D image (top) and point cloud (bottom) segmentation. From left to right, we show semantic, instance and panoptic segmentation.

defined for each pixel in the image while the 3D segmentation provides point-wise labels on the point cloud. We provide segmentation labels on 23 classes such as vehicle, pedestrian, vegetation, building, road, sidewalk, wall, traffic sign, pole and fence. Our segmentation labels can support a range of tasks such as image segmentation, video object segmentation, point cloud segmentation, multi-object tracking and segmentation (MOTS) [54] and multi-object panoptic tracking (MOPT) [19].

**Other labels.** In addition to above mainstream annotations, we also provide: (1) motion data for all agents including linear velocity, acceleration, and angular velocity. These motion data can be useful to ego-motion estimation, velocity estimation, tracking; (2) Fine-grained object class labels such as vehicle model class of Audi A2, Toyota Prius and Tesla Model 3; (3) Vehicle control signals such as throttle, steer, brake, and reverse; (4) City map and road structure, which is useful to localization, odometry and trajectory forecasting. Also, our dataset with point clouds and depth images can be used for point cloud forecasting [59] and depth estimation [32]. See supp. for details of other annotations.

### 3.3 High environmental variations

To learn perception systems robust to rare driving scenarios, it is important to first include lots of out-of-distribution data in the dataset for training and evaluation. However, collecting such data is difficult in the real world because they rarely happen and can be dangerous or at a high cost, especially for car crash. We leverage the simulator to intentionally generate such rare data and increase our environmental variations. We compare the environmental variations

Table 4: Comparison of environmental variations.

| Dataset | Adv. wea./light. | Crowded | High-speed | Vio. of rule | Crash |
|---|---|---|---|---|---|
| KITTI [12] | | | | | |
| Cityscape [9] | | | | | |
| Mapillary Vistas [35] | ✓ | | | | |
| ApolloScape [17, 55] | ✓ | | | | |
| SYNTHIA [44] | ✓ | | | | |
| H3D [38] | | ✓ | | | |
| SemanticKITTI [1] | | | | | |
| DrivingStereo [52] | ✓ | | | | |
| Argoverse [8] | | ✓ | | | |
| EuroCity [4] | ✓ | | | | |
| CADC [41] | ✓ | ✓ | | | |
| Audi [13] | ✓ | | | | |
| nuScenes [7] | ✓ | ✓ | | | |
| A*3D [40] | ✓ | | ✓ | | |
| Waymo Open [53] | ✓ | ✓ | | | |
| **Ours** (AIODrive) | ✓ | ✓ | ✓ | ✓ | ✓ |

between datasets in Table 4. Though recent datasets often have adverse weather/lighting conditions, some are limited by having too few number of agents. Also, existing datasets often collect data with ego-car driving at a low speed and barely have data of violation of traffic rules, let alone car crash. Instead, our dataset contains these rare data and has the highest environmental variations.

**Crowded scenes.** To learn perception systems robust to crowd, datasets with highly crowded scenes are needed. To that end, we collect many scenes with a high agent density. On average, we have 104 agents per frame within the sensing range. We show comparison of agents per frame and total labeled instances between datasets in Figure 6 (a). Note that some datasets such as KITTI and Cityscape have a relatively lower number of labeled instances because only objects in front are labeled.

**High-speed driving.** To mimic our daily driving speed, *i.e.*, 20 to 60km/h on local road and 80 to 120km/h on highway, we collect data by driving our ego-vehicle at a higher speed as shown in Figure 6 (b). Specifically, our driving speed has a wider distribution, ranging from 0 to 130 km/h.

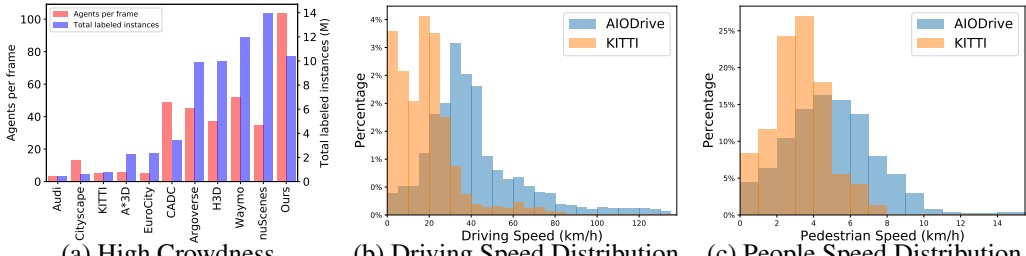

| (a) High Crowdness | (b) Driving Speed Distribution | (c) People Speed Distribution |

Figure 6: **Data Statistics**: (a) We compare agents density, which shows that our dataset has more crowded scenes; (b)(c) We compare the speed of ego-vehicle and pedestrians, showing that our data has wider distribution of speed including highway driving, person jogging and running.

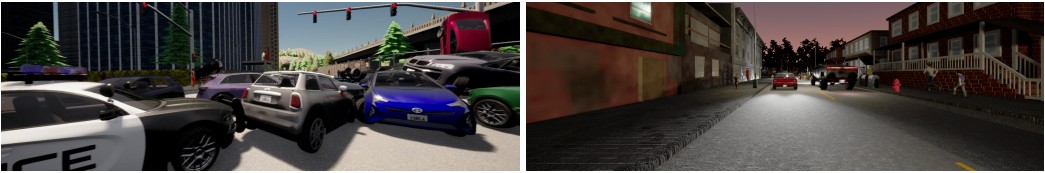

Figure 7: **Other Rare Data**. (Left): Car crash and piled up on highway. (Right): Driving at night.

**Other rare data.** We also provide adverse weather and lighting (*e.g.*, rainy, foggy and night. See Fig. 7 right for night), car crash (Fig. 7 left), vehicles that run over the red light, speed over the limit and aggressive lane changing, children and adults jogging and running. Though these data happens in the real world, they barely exist in existing datasets. To build robust perception systems, it is important to include these rare scenarios in the dataset. As an example, we show the pedestrian speed in Figure 6 (c), which contains jogging and running people. See supp. for details of other variations.

## 4 Experiments

To enable comparison with future work, we benchmarked baselines for a range of tasks including 2D detection, 3D detection, trajectory forecasting and point cloud forecasting[1]. Benchmarking for other tasks will be added. For fair comparison, annotation on the test set remains private while sensor data on train/val/test and annotation on train/val have been released. Please refer to supp. for data split.

### 4.1 2D object detection

We use FPN [28] with a ResNet50 [15] backbone as the baseline, where the backbone is pre-trained on ImageNet [10] and COCO [29]. We then fine-tune the baseline on AIODrive. The results are shown in the 1st row of Table 5, measured by the mean Average Precision (mAP) metric. Please refer to supp. for detailed detection evaluation protocol. We can see that FPN's performance is reasonable but lower than its performance on KITTI, *e.g.*, 93.53/89.35/79.35 for car in the easy/moderate/hard level. We believe this is because: (1) our evaluation requires detection at a larger range (more difficult) than KITTI, e.g., our 'hard' level requires detection of objects up to 120 meters while KITTI 'hard' level requires detection up to 70 meters; (2) AIODrive has a much higher object density than KITTI. As a result, there will be more occluded objects in the images which are hard to detect. With the challenges of long-range detection and detection in crowded scenes, we hope that our dataset can encourage future work to further push performance.

### 4.2 3D object detection

**Baselines.** We use LiDAR-based 3D object detection methods such as PointRCNN [49], PointPillars [21], SECOND [63] as baselines. See supp. for implementation details.

**Results on AIODrive with depth point clouds.** To reach the best performance, we first use our densest depth point cloud as inputs to baselines. As our point clouds have a longer range than prior datasets such as KITTI, we change the input point cloud range of detectors from 0-70m in frontal direction used in KITTI to 120m for all directions, to enable perception at a larger range.

Results are summarized in Table 5, where 3D detection performance is measured by mAP. Please refer to supp. for detection evaluation protocol. We can see that all 3D detection baselines achieve

---

[1]The baseline and evaluation code have been released at `https://github.com/xinshuoweng/AIODrive` for users to reproduce baseline results and evaluate future methods.

Table 5: Quantitative results of 2D/3D object detection baselines on the AIODrive test set.

| Method | Input Data | Output Modalities | Car | | | Pedestrian | | | Cyclist | | |
|---|---|---|---|---|---|---|---|---|---|---|---|
| | | | Easy | Moderate | Hard | Easy | Moderate | Hard | Easy | Moderate | Hard |
| FPN [28] | RGB from 5 cameras | 2D | 89.45 | 78.66 | 69.51 | 92.88 | 87.28 | 75.50 | 94.15 | 90.80 | 72.10 |
| PointRCNN [49] | Depth point cloud | 3D | 78.13 | 77.99 | 73.63 | 58.73 | 53.71 | 44.74 | 59.03 | 53.85 | 49.36 |
| PointPillars [21] | | | 80.86 | 77.39 | 69.77 | 55.37 | 47.79 | 40.94 | 60.72 | 50.20 | 46.35 |
| SECOND [63] | | | 81.35 | 79.38 | 70.57 | 62.32 | 59.23 | 54.34 | 61.45 | 58.49 | 52.86 |

Table 6: 3D detection results using point cloud with different densities in our AIODrive dataset.

| Method | Point Density (# of points) | Car | | | Pedestrian | | | Cyclist | | |
|---|---|---|---|---|---|---|---|---|---|---|
| | | Easy | Moderate | Hard | Easy | Moderate | Hard | Easy | Moderate | Hard |
| PointRCNN [49] | 100,000 (Velodyne-64 LiDAR p.c.) | 74.98 | 72.73 | 53.85 | 45.31 | 37.37 | 34.66 | 56.95 | 50.70 | 42.96 |
| | 600,000 (Long-range LiDAR p.c.) | 76.74 | 75.17 | 69.76 | 56.39 | 50.14 | 40.38 | 58.71 | 52.37 | 46.83 |
| | 1,000,000 (Long-range LiDAR p.c.) | 77.71 | 77.26 | 71.17 | 58.16 | 51.92 | 43.81 | 59.64 | 52.61 | 47.73 |
| | 4,000,000 (Depth p.c.) | 78.13 | 77.99 | 73.63 | 58.73 | 53.71 | 44.74 | 59.03 | 53.85 | 49.36 |
| | 1,000,000 (SPAD-LiDAR p.c.) | 77.83 | 71.41 | 63.30 | 59.88 | 53.43 | 44.79 | 61.10 | 55.69 | 48.80 |

reasonable performance on our AIODrive dataset. Also, performance tends to decrease significantly from the 'easy' to the 'moderate' and then to the 'hard' level where the required detection range is increasing (see supp. for detailed evaluation protocol). Again, this shows that detection at a longer range is harder than detection of nearby objects. We hope that our high-density long-range point clouds can be used to encourage future research towards improving long-range 3D object detection.

**Effect of point cloud density.** To show usefulness of our high-density point clouds, now we evaluate the same detector using point clouds with different density levels. Also, we adapt PointRCNN and show the first 3D detection baseline that works with SPAD-LiDAR point cloud inputs. We summarize the results in Table 6. We can see that, using (LiDAR and depth) point clouds with a higher density as input generally achieves higher performance, especially in the 'hard' level which includes faraway objects up to 120m. This suggests that high-density long-range point clouds could be helpful for improving 3D detection at a longer range. Also, for LiDAR and depth point clouds with different densities, we found that the differences of performance in the 'easy' level are not significant (except for pedestrians). This shows that, for cars and cyclists, the main performance bottleneck of 3D detection at nearby range (up to 40 meters in the 'easy' level) may not be point cloud density but other factors such as model capacity. In contrast, detection for nearby pedestrians can be significantly improved using point clouds with a higher density.

We also observed a different performance pattern when using SPAD-LiDAR (the last row in Table 6), which tends to achieve higher performance for pedestrians and cyclists (small objects) and lower performance for cars (large objects). We hypothesize that the higher performance for small objects may be due to the larger fill factor of the SPAD-LiDAR compared to APD-LiDAR (see supp. for details about fill factor). However, it is not fully clear why performance drops for cars. We hypothesize that it is because our method of using SPAD-LiDAR by merging multiple point cloud returns (see supp. for implementation details) does not fully exploit multi-echo information in the raw 3D tensor data. Future work is needed to fully leverage the SPAD-LiDAR data for 3D detection.

**Results on real-world KITTI data.** Lastly but also importantly, we investigate if using our dataset can improve performance on the real data. To that end, we augment the KITTI training data with the data from our dataset to train PointRCNN [49]. This data augmentation is achieved by equally (same number of frames) combining data from two datasets in every batch of training. In the case we have a total of more frames from AIODrive than KITTI, we randomly sample frames from AIODrive and still maintain an equal number of frames from two datasets in every batch. We follow the KITTI evaluation on the test set and summarize the results in Table 7. We can see that PointRCNN trained with only KITTI data (the 2nd row) achieves similar performance for car as reported in [49]. Also, PointRCNN trained with only synthetic AIODrive data (the 1st row) achieves lower performance on KITTI compared to trained with the KITTI data. This suggests that domain gap exists between two datasets. Importantly, when we augment training data by combining data from two datasets (the 3rd and 4th rows), we observed clear performance improvements. This proves that our AIODrive data can be used in concert with real data to improve performance on the real data. Moreover, higher performance is achieved if more augmented frames (*e.g.*, all frames vs. 10k frames) are used. The best performance is achieved when both KITTI and all data from AIODrive are used for training.

### 4.3  Trajectory forecasting

**Baselines.** In addition to benchmark 2D and 3D object detection, which depend on only the object box annotation, we also benchmark trajectory forecasting to understand how challenging the trajectory

Table 7: 3D detection results on the KITTI dataset when training is augmented with AIODrive data.

| Method | Training Data | Car | | | Pedestrian | | | Cyclist | | |
|---|---|---|---|---|---|---|---|---|---|---|
| | | Easy | Moderate | Hard | Easy | Moderate | Hard | Easy | Moderate | Hard |
| PointRCNN [49] | AIODrive | 65.32 | 46.21 | 39.38 | 24.57 | 19.04 | 18.32 | 40.93 | 30.41 | 26.68 |
| | KITTI | 85.02 | 75.16 | 68.14 | 46.53 | 38.76 | 33.96 | 73.40 | 56.73 | 51.87 |
| | KITTI + AIODrive 10k frames | 87.24 | 76.83 | 70.53 | 46.97 | 40.78 | 36.03 | 74.19 | 59.31 | 52.93 |
| | KITTI + AIODrive all frames | 88.10 | 77.03 | 72.41 | 51.03 | 42.18 | 37.26 | 78.01 | 60.14 | 52.89 |

Table 8: Quantitative results of trajectory forecasting baselines on the AIODrive test set.

| Method | Pred. 20 frames (2s) | | | | | | Pred. 50 frames (5s) | | | | | |
|---|---|---|---|---|---|---|---|---|---|---|---|---|
| | ADE↓ | FDE↓ | SADE↓ | SFDE↓ | APD↑ | FPD↑ | ADE↓ | FDE↓ | SADE↓ | SFDE↓ | APD↑ | FPD↑ |
| Social-GAN, Car | 1.263 | 2.293 | 1.727 | 3.475 | 5.074 | 10.971 | 4.304 | 6.564 | 5.600 | 9.464 | 10.546 | 19.942 |
| Social-GAN, Pedestrian | 1.258 | 2.172 | 1.826 | 3.534 | 2.070 | 4.135 | 3.308 | 5.448 | 4.602 | 8.276 | 4.275 | 8.849 |
| Social-GAN, Cyclist | 1.420 | 2.656 | 1.619 | 3.292 | 9.571 | 21.122 | 4.393 | 7.284 | 4.895 | 9.006 | 13.005 | 25.851 |
| Social-GAN, Motorcycle | 1.828 | 3.310 | 2.223 | 4.402 | 7.218 | 15.225 | 5.375 | 8.415 | 6.525 | 10.902 | 19.721 | 37.772 |
| Social-GAN, Average | 1.442 | 2.608 | 1.858 | 3.676 | 5.983 | 12.863 | 4.345 | 6.928 | 5.405 | 9.412 | 11.887 | 23.104 |
| AgentFormer, Car | 0.876 | 1.408 | 1.549 | 3.071 | 4.976 | 10.818 | 2.349 | 3.094 | 4.311 | 7.835 | 10.913 | 20.170 |
| AgentFormer, Pedestrian | 0.798 | 1.167 | 1.708 | 3.268 | 3.455 | 6.908 | 1.893 | 2.565 | 4.314 | 7.983 | 8.648 | 16.776 |
| AgentFormer, Cyclist | 1.302 | 2.177 | 1.515 | 3.065 | 4.280 | 7.531 | 2.621 | 3.952 | 2.918 | 5.539 | 5.598 | 11.609 |
| AgentFormer, Motorcycle | 1.730 | 2.603 | 2.709 | 5.024 | 7.388 | 13.492 | 3.547 | 4.580 | 5.061 | 8.311 | 8.374 | 16.551 |
| AgentFormer, Average | 1.176 | 1.839 | 1.885 | 3.607 | 5.025 | 9.687 | 2.602 | 3.547 | 4.151 | 7.417 | 8.383 | 16.277 |

data is in the AIODrive dataset. We use the most popular method Social-GAN [14] as our baseline. Also, as Social-GAN is relatively outdated so we benchmark another recent state-of-the-art approach AgentFormer [66]. Please refer to instruction page for detailed evaluation protocol.

**Metrics.** We use standard ADE/FDE (Average/Final Displacement Error), and also SADE/SFDE (Scene-specific ADE/FDE), APD/FPD (Average/Final Pairwise Distance). Please refer to instruction page for detailed explanation of each metric. In brief, ADE/FDE are used to measure prediction accuracy for each agent individually while SADE/SFDE are used to measure prediction accuracy for all agents in the scene jointly. Also, APD/FPD are used to measure diversity of generated trajectories.

**Results.** We summarize the results in Table 8. Overall, both methods perform reasonably considering challenging out-of-distribution trajectories are present in the AIODrive dataset, *e.g.*, complex interaction, car crash. Moreover, AgentFormer consistently outperforms Social-GAN in terms of accuracy (for each object category or on average), similar to the performance trend of two methods on other datasets (*e.g.*, ETH/UCY [39, 22], nuScenes [7]).

## 4.4  Point cloud forecasting

**Baselines.** As a new task in autonomous driving, we currently do not have many publicly available baselines except for SPFNet [60]. Also, we create one variant as a stronger baseline for benchmarking in addition to the original SPFNet. Specifically, we replace the 1D-LSTM used in SPFNet with Conv-LSTM for better feature learning. We use 100k-point LiDAR data for both baselines.

**Metrics.** Following the evaluation protocol in [60], we use standard Chamfer distance (CD) and Earth mover's distance (EMD) to measure accuracy of predicted point clouds compared to ground truth point clouds. Also, we evaluate prediction horizon of 1 and 3 seconds.

**Results** are summarized in Table 9. We found that performance of both baselines is in the reasonable range of CD and EMD, although EMD are higher than in KITTI as reported in [60]. We believe this is because AIODrive dataset has much higher object density compared to KITTI so it is more challenging for point cloud forecasting methods to deal with complex object motions and predict correct object locations. We hope that this high object density challenge can encourage future research. Meanwhile, as CD are generally dominated by global point cloud structures (*e.g.*, road, building) and AIODrive 100k-point LiDAR is designed to be similar to KITTI velodyne-64, CD errors are at a similar level in AIODrive and KITTI.

Table 9: Point cloud forecasting benchmarking.

| Method | Pred. 10 frames (1s) | | Pred. 30 frames (3s) | |
|---|---|---|---|---|
| | CD↓ | EMD↓ | CD↓ | EMD↓ |
| SPFNet [60] | 0.838 | 438.499 | 0.852 | 446.593 |
| SPFNet-ConvLSTM | 0.507 | 366.985 | 0.554 | 376.208 |

## 5  Conclusion

We proposed a dataset with the most diverse annotations, environmental variations and sensors. Our dataset can support all mainstream perception tasks and innovate multi-task multi-sensor perception systems. Also, we confirmed that our high-density long-range point clouds can be used to improve long-range perception. To enable public comparison and encourage future research in long-range perception, our full dataset and accompanying code will be released.

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
