# OpenReview forum: "All-In-One Drive: A Comprehensive Perception Dataset with High-Density Long-Range Point Clouds"
_NeurIPS.cc/2021/Track/Datasets_and_Benchmarks/Round1 — Submitted to NeurIPS 2021 Datasets and Benchmarks Track (Round 1)_

### Official Review · Reviewer_rm7y · 2021-06-20
**The usefulness of a synthetic dataset for autonomous driving is limited**

**Rating:** 5
**Confidence:** 5
**Correctness:** The claims are correct, and the datas…
**Clarity:** The paper is well-written.

**Strengths:**

This synthetic dataset contains:

(1)  a comprehensive sensor suite (RGB/Stereo/Depth cameras, LiDAR, SPAD-LiDAR, Radar, IMU, GPS).

(2)  annotations for multiple perception tasks (detection, tracking, prediction, segmentation, etc.).

(3)  out-of-distribution driving scenarios (vehicle crash, violation of traffic rules, etc.)

**Weaknesses:**

I updated my rating to 5. Please refer to the reasons.

(1) The usefulness of a synthetic dataset is limited. As is mentioned in L.78-81, the synthetic data cannot be used independently and should only be used in concert with real-world data, which restricts its potential application scope. Directly adapting models trained on synthetic data to real-world scenarios leads to a significant performance drop (39.38 compared to 68.34 of the baseline on KITTI in Table 7).

(2) Whether synthetic data can really help real-world tasks should be further validated. Table 7 only shows the synthetic data improves the detection results of a lower baseline (PointRCNN) on a small amount of real data (KITTI). It will be more convincing if the performance of a state-of-the-art detector (e.g. PV-RCNN) can be improved on a large-scale dataset (Waymo) with synthetic data. The effects on other tasks (e.g. segmentation, tracking) should also be evaluated.

(3) The dataset size is not large. It only contains 100k frames of point clouds, compared to 230k frames in the real-world Waymo dataset. I believe the post-processing and storage issue mentioned in appendix 5 can be addressed by code optimization and compression since other datasets can also overcome similar issues.

(4) The whole data is generated by the widely-used CARLA simulator without many modifications. Thus it can only be considered as an extension of the existing simulation platform instead of a new data source. Efforts that can improve the current data generation pipeline and make the generated data look more real will be appreciated.

(5) Some crucial experiments are missing. As is mentioned in L.201-207, 2D and 3D segmentation labels are provided, but I didn't find any experiments or baseline results on both 2D and 3D segmentation in the paper and appendix.

**Additional Feedback:**

Here are my suggestions:

(1) The authors should provide baseline results on 2D and 3D segmentation, and validate whether synthetic data can improve real-world segmentation tasks on CityScapes and SemanticKITTI.

(2) The authors should provide more comprehensive experiments to validate the effectiveness of using synthetic data. For example, try training PV-RCNN with the data from both Waymo and AIODrive and compare it with the performance on Waymo.

(3) It will be very interesting if some modifications are proposed to improve the CARLA simulator and make the synthetic data look more real. I suggest the authors take a look at SurfelGAN (https://arxiv.org/abs/2005.03844) and LiDARsim (https://arxiv.org/abs/2006.09348). Directly using CARLA is quite trivial.

**Documentation:**

The paper contains sufficient details on data collection and organization, availability and maintenance, and ethical and responsible use.

**Ethics:**

There is no ethical concern since the data is synthesized.

**Relation To Prior Work:**

The paper can be viewed as an application of the CARLA simulator. The descriptions of data generation are detailed, but there is no significant modification to improve the generated data quality of CARLA.

**Summary And Contributions:**

This paper provides a comprehensive perception dataset for autonomous driving. The data is generated by the widely-used CARLA simulator.

---

### Official Review · Reviewer_n2h6 · 2021-07-02
**The size of the dataset is not competitive considering that this is a synthetic data**

**Rating:** 6
**Confidence:** 4

**Strengths:**

The main strength of the dataset are diverse annotations, the all-in-one nature of the dataset could promote multimodal research in the field.

The authors simulate a long-range lidar sensor and SPAD-Lidar sensor, which are not deployed in real cars but still in development. This can be seen as both a strength and a weakness, on one hand the research using these high precision sensors is necessary for the next-generation cars, on the other hand, for foreseeable future the models build using the synthetic dataset and provisory sensors can't be tested on real data.

Many scenes difficult to obtain in real-world datasets are provided here.


**Weaknesses:**

After reading the authors response, I change my rating to 6 and agree that the dataset is sufficiently large.
I believe that the proposed dataset introduces novelty and will be useful for the community. I hope that the authors provide the additional benchmarks as noted by R1 and R3.

_______________________________________________________________________________



The number of annotated frames in the dataset is lower (roughly by half) than in recently proposed, real-world datasets [52, 53]. One of the main advantages of synthetic datasets is the automatic and free data collection. The authors didn't take the full advantage of this fact. The experiment show in Table 7 shows that using synthetic dataset is beneficial in real-world evaluation settings. They observe ~2% boost in hard cases of detecting cars when using 10% or 100% of frames. We don't know if the improvement would saturate when using more synthetic data.
The authors claim that it's easier for the users to download smaller datasets, but they could just upload subsets of data to facilitate download process.

The main novelty in this dataset are the diverse sensors and new generation LIDAR sensors. Proposed simulated sensors are not deployed in real-world. While the annotations they provide are more accurate, this line of research can't be (at the moment) tested in real-world.

**Additional Feedback:**

I believe that this dataset will be beneficial for the community, however, given that the dataset was generated synthetically, its size should be comparable to recent, real-world datasets. The cost of storage is minimal compared to large-scale data collection of any vision datasets. This track offers two submission deadlines, should the authors decide to submit to the later one, the collection process could have continued.

Additionally, please see my comments in the Documentation section.

**Clarity:**

Overall the paper is well written and easy to follow. Please re-write the sentence in lines 176-177 for clarity.
There are minor errors that should be corrected in the final version eg:
177: and 1km range. > and a 1km range
178: a car at 130 meters, depth point cloud> a car at 130 meters, a depth point cloud

**Correctness:**

The dataset is constructed in the correct way. The authors provide benchmarks on 4 perception tasks, using multiple methods.

**Documentation:**

There is sufficient information on the data collection, availability, and ethical use. The authors are not confident in how long the data will be hosted on their server (278-279 supp).

The clear documentation of the out-of-distribution scenes in the dataset is missing. A clean table showing the percentage of those sequences or characterization of the sequences with different attributes would be helpful for a user. Also, it would be helpful to see how many of such sequences are in the training/validation/test set. The sequences could potentially contain a label of attributes to facilitate the research.


**Ethics:**

The dataset was created in a simulated environment, the data collection was automatic and neither human annotators nor human actors were involved.

Theoretically, the presented scenes of dangerous events could be used to train models adversely. However, those examples are also necessary to train robust perception models.

**Relation To Prior Work:**

The paper discusses previous automonous driving datasets and differences between them.


**Summary And Contributions:**

The paper presents a synthetic dataset of driving scenarios, the dataset was generated using an open-source simulator CARLA. The main contributions of the proposed dataset are annotations from diverse sensors, and rare, out-of-distribution driving scenes.

The authors generated 100k frames with eight modalities including: RGB, Stereo, Depth, 10 LiDAR, SPAD-LiDAR, Radar, IMU, GPS. The LIDAR sensors used for generating the annotations provide higher density / multi-echo point clouds.

Due to synthetic nature of the simulator, the authors are able to safely generate images of dangerous events (car crashes, violation of traffic rules), difficult driving scenarios (bad weather conditions), and crowded scenes.

---

### Official Review · Reviewer_2ikT · 2021-07-05
**A synthetic multi-modal dataset/benchmark for existing self-driving perception tasks based on CARLA**

**Rating:** 6
**Confidence:** 4
**Clarity:** Yes. But please see W3.

**Strengths:**

S1. The synthetic dataset is comprehensive in terms of modalities, tasks, and data distribution, and can be a good supplement for the real-world dataset as the pre-train dataset or the test set of out-of-distribution scenarios.

S2. The high-density and long-range sensor is an important complement of existing sensing modalities which is not widely available to the research community yet, so this dataset can promote the development of long-range sensing in autonomous driving.

S3. The dataset is open-source, and the website is professional. Perhaps it can attract researchers to use the dataset due to its comprehensiveness.

**Weaknesses:**

W1. Its comprehensiveness is mainly because the dataset is totally synthetic, and it is convenient to increase the modalities, annotations, and data distributions in the simulation. Thus it seems that the novelty of this work should mainly be attributed to CARLA rather than the dataset itself, although this reviewer acknowledged the hard work of the authors for dataset generation.
It would be more exciting for the reviewer to see some new self-driving tasks proposed along with this dataset.

W2. In addition to the domain gap mentioned, the simulated data have more differences than the real-world data, e.g., the authors did not mention the synchronization issue when generating multi-modality data, i.e., is it reasonable in the real world that all the sensors are recorded at the same frequency? Moreover, there is no discussion of the performance of the perception system under various scenarios like adverse weather, different speed, etc.

W3. The organization can be further improved because the dataset part is a bit wordy yet the benchmark part is not comprehensive since the tracking and segmentation tasks are not included. Maybe the authors can condense the dataset part to leave more space for the benchmark.

W4. Not all the baseline approaches are open-sourced, for example, there is no code for SPFNet [60] even in the repository of [60]. The authors should provide all the baseline code as well as the evaluation code to guarantee that the benchmark can work well.

W5. Actually, it is not fair to compare the synthetic dataset with the real-world dataset in terms of comprehensiveness such as in Table 1 and Table 3. The authors should not overclaim their contribution since real-world works are much more complex and difficult than synthetic works. It is better to claim that this dataset can be a good supplement to the real-world existing dataset, but not that it has included all the attributes of the existing dataset.

**Additional Feedback:**

No. Please see the mentioned weaknesses.

**Correctness:**

It is difficult for the reviewer to evaluate the correctness comprehensively but everything in the manuscript looks normal, and the authors promised that they will open-source the baseline codes.

Also, please see W2.

**Documentation:**

Yes. The website looks comprehensive and professional.

**Ethics:**

I do not see any ethical concerns for this dataset.

**Relation To Prior Work:**

Yes. But please see W5.

**Summary And Contributions:**

This work proposed a synthetic comprehensive perception dataset in autonomous driving scenarios based on CARLA. The main contribution lies in the comprehensiveness in terms of modalities, tasks as well as distributions. Moreover, the authors developed high-density long-range sensors based on the simulator, to enable research in long-range perception. Overall, this work is well-written, clearly motivated, and can be combined with real-world datasets to promote the development of self-driving. However, the dataset is totally synthetic, and the benchmark has not considered tracking as well as segmentation.

---

### Comment · Reviewer_rm7y · 2021-07-18
**Some explanations on the final rating changes**

Before the rebuttal, I gave my rating as 6, mainly because I believe the authors could add the missing baselines and experiments in the rebuttal. However, I don't think the added baselines on 3D MoT and segmentation are sufficient, since they didn't provide baselines on 2D segmentation they plan to support in the paper. And I really believe more experiments on whether this scale of synthetic data (100k frames) could really help the real-world tasks (e.g. 230k frames of 3D detection in Waymo) should be further conducted. Given the above reasons, I think the current experiments are far from satisfactory. Thus I change my rating to 5.

I appreciate the authors' efforts in building this synthetic dataset, and I think this synthetic dataset may be promising to the autonomous driving community. However, I cannot agree with the authors' response that "requesting more experiments is unfair and not reasonable". Here are my reasons:

(1) The authors' claim that their dataset is a  "comprehensive perception dataset" in their paper title cannot be well supported if they cannot provide baselines for every task they proposed. I understand the authors have a heavier workload and have provided more baselines compared to Waymo/nuScenes/Lyft. However, Waymo/nuScenes didn't claim that they are "comprehensive perception datasets", so it is not necessary for them to provide more baselines. I think the comprehensiveness of this synthetic dataset is the most significant contribution in the paper, and it is the only important advantage I think compared to the existing large-scale real-world AD datasets. After the rebuttal, baselines on 2D segmentation are still missing. Thus I think the existing experiments cannot support their goals and the authors overclaimed their contributions.

(2) The dataset part and the experiments part are not consistent in the paper. The paper has a chapter called "2D-3D Segmentation" in L.201-207 and they provide a figure of annotation in Figure 5, but the authors didn't provide any baselines results on the two tasks before the rebuttal. Even after the rebuttal, the experiments on 2D segmentation are still missing. It looks so strange that the authors discussed and promised their dataset supports 2D/3D segmentation in the paper but they didn't provide baselines in the experiments part.

(3) I still think the experiments on whether the synthetic data could help the real-world tasks are of high priority and really significant. I don't think the results in Table 7 are convincing, because KITTI is a small dataset compared to AIODrive, and PointRCNN is a relatively low baseline. As a researcher also in this area, honestly speaking I can train PointRCNN with the official code to attain much higher performance  (78.5 mod. AP) using only KITTI data, even higher than the authors' provided results (77.03) using both KITTI and the synthetic data. Thus before the rebuttal, I suggest the authors try Waymo which is larger than their proposed AIODrive, and use a state-of-the-art 3D detector, but the authors didn't provide the respective results even after the rebuttal. The usefulness of this synthetic dataset is uncertain without those crucial experiments.

Overall, I think the current experiments are far from satisfactory. If the paper is accepted, there is no guarantee that the authors will continue to provide the crucial experimental results. And without further validation, there is a chance that the researchers may find this synthetic dataset is not useful to real-world autonomous driving tasks, which could potentially have a negative effect on the community. In conclusion, I think the authors should spend more time perfecting their experiments and turn to the 2nd round.

---

### Decision · Program_Chairs · 2021-07-26

**Decision:**

Reject

**Comment:**

The rating of the paper has been fluctuating between 5 and 6, reflecting the struggle of balancing the merits and drawbacks of the paper. While we value the contribution of datasets in this special NeurIPS track, we expect the dataset papers can demonstrate distinction from the existing ones and provide arguments about the potential new findings on the proposed dataset. However, the reviewers expressed concerns about the practical usage scenarios of the proposed dataset. While a dataset will a comprehensive set of annotations and tasks can be very useful, the advantage of the comprehensiveness is not well studied in the paper. In addition, our reviewers, who are experienced in the related tasks and models, raised concerns about the quality and quantity of the provided baseline models on the new datasets. In the end, we conclude that while the proposed dataset has good potential in advancing machine learning research, it is not ready for acceptance yet. We encourage the authors to revise the papers based on the review feedback.